# Attitude and knowledge of healthcare providers towards female genital cosmetic surgery: A cross-sectional study

Zinat Ghanbari[1], Saeedeh Shirdel[1], Mohadese Dashtkoohi[1], Ideh Rokhzadi[1], Misa Naghdipour Mirsadeghi[2], Mahroo Rezaeinejad[1], Marjan Ghaemi[1]*

1 Vali-E-Asr Reproductive Health Research Center, Family Health Research Institute, Imam Khomeini Hospital Complex, Tehran University of Medical Sciences, Tehran, Iran, 2 Department of Gynecology, Reproductive Health Research Center, Guilan University of Medical Sciences, Guilan, Iran

* marjan_ghaemi@yahoo.com

## Abstract

### Background

Female genital cosmetic surgery (FGCS) has experienced increasing popularity over the past decades. This procedure includes surgical and non-surgical intervention to alter genital appearance, often without medical indications. As many women are unfamiliar with normal genital anatomy, healthcare providers play a critical role in guiding patients. Therefore, evaluating their attitudes and knowledge for identifying gaps in training and ethical issues could be valuable.

### Materials and methods

This cross-sectional study was conducted from 17 January 2023 to August 2024 among 320 healthcare providers in Iran. Participants included obstetricians and gynecologists, plastic surgeons, midwives, gynecology residents, urologists, and general practitioners. Data were collected using a validated web-based questionnaire about participants' demographics, clinical experience, knowledge of FGCS, risk awareness, ethical considerations, and attitudes toward FGCS. SPSS version 27 was used for statistical analysis.

### Results

The majority of participants (57.6%) were obstetricians and gynecologists. Most respondents reported being approached by patients seeking FGCS, with body image dissatisfaction (65.4%) and partner requests (62.2%) being the primary complaints. Awareness of short-term risks varied significantly across specialties (p < 0.001), with obstetricians and gynecologists being the most informed. Confidence in performing procedures was highest among plastic surgeons, while ethical considerations were more recognized among obstetricians and gynecologists.

**Data availability statement:** Due to ethical restrictions imposed by the Institutional Review Board (IRB) of Tehran University of Medical Sciences and ValiEAsr Reproductive Health Research Center in accordance with Iranian national biomedical research ethics guidelines overseen by the National Committee for Ethics in Biomedical Research (Ministry of Health and Medical Education), sharing the de-identified dataset is not permitted. The data contain potentially sensitive information related to patients' attitudes toward intimate gynecological procedures, which could lead to indirect identification despite de-identification efforts, given the specialized nature of the study population (Iranian gynecology specialists in a limited geographic area). Data access requests should be directed to the head of the Family Health Research Center Professor Sedigheh Hantoushzadeh email: hantoushzadeh@ tums. ac.ir. Contact: 00989121271209 Alternatively, contact the National Committee for Ethics in Biomedical Research: Tel: +98-21-81455618 Address: Floor 13, Block A, Ministry of Health & Medical Education Headquarters, Qods Town, Tehran, Iran.

**Funding:** The author(s) received no specific funding for this work.

**Competing interests:** The authors have declared that no competing interests exist.

Although most participants in all groups (68.6% of OB/GYN, 90.0% of plastic surgeons and 75.2% of others) reported formal training, long-term risk awareness and procedural confidence gaps persisted. Only a minority endorsed FGCS for purely aesthetic reasons (8.6% of obstetricians and gynecologists, and 1.6% of others). More than half of respondents (56.8% of obstetricians and gynecologists, 60% of plastic surgeons, and 55.6% of others) agreed that all women should undergo consultation before FGCS.

## Conclusion

Healthcare providers demonstrated diverse levels of knowledge and confidence in FGCS, which is influenced by specialty and training. Attitudes towards performing procedures solely for aesthetic purposes, including for those under 18, were generally positive, although pre-procedure counseling was also emphasized. While ethical considerations were widely acknowledged, a targeted training program is needed to address gaps in risk awareness and improve providers' confidence.

## Introduction

Female genital cosmetic surgeries (FGCS) have experienced a remarkable increase in demand over the past few decades. FGCS refers to the surgical or non-surgical alteration of female genitalia, usually for aesthetic purposes [1]. These include labiaplasty, hymenoplasty (revirgination surgery), vaginoplasty (tightening or restoring vaginal contour), perineoplasty, clitoral hood reduction (removal of excessive tissue around the clitoris), monsplasty (mons pubis reconstruction and reduction), G spot augmentation (injecting hyaluronan into the anterior vaginal wall to augmentation of the G-spot in women.), frenuloplasty, and fat or filler injections, especially to labia majora and mons pubis. Among these, labiaplasty is the most common [2].

According to the Aesthetic Society's Cosmetic Surgery National Data Bank, despite the restrictions caused by the COVID-19 pandemic, in 2020, total cosmetic surgery revenue exceeded US$9 billion. At the same time, 13,697 labiaplasties have been performed [3]. This number shows an increasing trend compared to previous years: 12,903 and 9,945 labiaplasties were conducted in 2019 and 2015, respectively [4].

One of the main reasons for this growing rate may be sociocultural norms that shape women's expectations of normal genitalia [5,6]. The increased attention of women toward their genital area despite a lack of formal education about anatomy and variation of genitalia is also notable. The focus on the appearance and cleanliness of this area, including the removal of pubic hair through shaving, waxing, or permanent laser treatment, has drawn attention to the region [7,8]. Additionally, much information is obtained online. Unfortunately, adolescents and young women frequently use online platforms as an educational resource for exploring their bodies and sexual desires without supervision, leading them to encounter both

accurate and inaccurate information inevitably. The increasing prevalence of images depicting hairless genitalia with small labia in pornographic content has created a distorted perception of what is considered normal variation in the natural human body [9].

FGCS often present a complex combination of medical, cultural, ethical, and psychological considerations, posing challenges for healthcare providers in distinguishing between necessary medical intervention and optional cosmetic procedures, requiring them to navigate ethical dilemmas while respecting patient autonomy [10]. As medical staff are usually the first to discuss with women seeking FSGS, the decisions patients make are greatly influenced by their perspectives and knowledge. Research indicates that a significant number of healthcare providers may not have a comprehensive understanding of all these aspects. This lack of awareness could result in inconsistent guidance and treatment, potentially putting patients at risk [11].

Healthcare providers may hold diverse attitudes towards FGCS, influenced by their personal beliefs, professional training, and interpretation of medical standards, ultimately impacting the quality of patient care [12]. Medical ethics principles, including beneficence and nonmaleficence, raise questions about the risk of surgery to improve physical appearance without medical indication. Many critics are concerned about cosmetic surgery's social and cultural aspects [13]. Healthcare providers must also understand the potential risks and complications associated with FGCS. And thoroughly explain them to patients. The possible negative results include infection, the development of scars, wound dehiscence, permanent undesirable appearance, dyspareunia (painful sexual intercourse), changes in sexual sensation, reduced vaginal lubrication, and psychological distress. Patients deserve to be aware of the limited high-quality evidence supporting the effectiveness of genital cosmetic surgical procedures. [14–16]. By evaluating healthcare staff's knowledge, we can enhance their awareness and plan for their training. Although numerous studies examined women's motivation to seek FCGC, there is still limited data regarding healthcare provider attitudes. In a survey among medical students and specialists, most participants stated that FGCS should only be carried out at the patient's request, and more than half of them stated that it could enhance self-esteem, quality of life, and sexual function [17]. Another study demonstrated that most physicians in Saudi Arabia had positive attitudes toward FGCS [18].

This article aims to assess healthcare's knowledge and attitude toward FGCS. Through this exploration, the article seeks to enhance our comprehension of how healthcare professionals' approach FGCS and its impact on patient treatment. The discoveries could play a role in improving policies, refining healthcare provider education, and steering future FGCS research in contemporary healthcare.

## Materials and methods

This cross-sectional study was conducted in Iran from January 17, 2023, to August 5, 2024, involving physicians across diverse healthcare settings and midwives in urban healthcare facilities in Iran.

### Ethical approval and consent to participate

The ethics committee of Tehran University of Medical Science approved the study. (approval number: IR.TUMS.IKHC.REC.1401.202).

### Participant selection

Registered healthcare providers with a valid medical council number were randomly invited from a pool of randomly selected from eligible participants including obstetricians and gynecologists, plastic surgeons, midwives, gynecology residents, urologists, and general practitioners. The inclusion criteria were healthcare providers who have at least one year of professional experience, are willing to participate and provide informed consent, and have access to electronic communication and the Internet. Participants who missed more than 15% of questions were excluded.

## Questionnaire design and data gathering

a structured, web-based questionnaire was developed based on a comprehensive review of the literature and experts' consultation. The questionnaire comprised multiple-choice and Likert-scale items. A pilot version of the questionnaire was distributed to a sample of 20 healthcare providers to improve clarity. Content validity was evaluated by calculating a content validity index (CVI) based on expert feedback. Items that received a CVI score of ≤0.8 were either revised or eliminated. Construct validity was examined through Exploratory Factor Analysis (EFA) of pilot responses to confirm the measurement of intended domains. A test-retest approach was utilized for reliability assessment. The final version was distributed to 20 providers with a two-week interval, and the reliability of the questionnaire was assessed using Cronbach's alpha for internal consistency and interclass correlation coefficient (ICC). A Cronbach's alpha ≥ 0.7 was considered acceptable, while an ICC value > 0.75 indicates good test-retest reliability. A text message invitation was sent to the chosen healthcare providers, containing details about the study's goals and a link to the Google Form questionnaire. Information was gathered online and anonymously through Google Forms. The survey consisted of 20 questions in 4 different sections:

Baseline characteristics of participants (Age, gender, specialty, years of experience, and practice setting)

Knowledge and confidence (patients' complaints, source of knowledge, confidence, formal training)

Risk awareness and ethical practice (awareness of short-term and long-term risk, informed consent, and ethical issues)

Participant's attitude (attitude toward patient selection and FGCS, advertising, popularity of FGCS among other cosmetic procedures…). Advertising refers to public promotional activities such as online campaigns, clinic websites, and social media used by healthcare providers or clinics to promote FGCS services.

Additionally, this study focused exclusively on cisgender women

## Statistical analysis

SPSS statistical software version 27 was utilized for the data analysis. The calculation for sample size was carried out using a 95% confidence interval, a margin of error of 5%, and an expected response rate. Accounting for potential non-responders, the sample size of 300 was determined to be enough. The descriptive data were presented as number (n), prevalence (%), mean, and standard deviation. The chi-square test was used to compare qualitative data such as patients complains, source of information, and risk awareness. Meanwhile, the one-way ANOVA test was employed to compare numeric variables. A p value < 0.05 was considered statistically significant.

## Ethical consideration

This study was approved by the Ethical Committee of Tehran University of Medical Sciences (IR.TUMS.IKHC. REC.1401.202). were provided with comprehensive information about the study objectives through a text message. Written informed consent was obtained from all individual participants included in the study. Anonymity was rigorously maintained throughout the data collection process to protect the privacy of participants.

## Results

320 participants from different departments were included in the study, including obstetricians and gynecologists, plastic surgeons, and midwives. gynecology residents, urologists, general practitioners, and others. Respondents were divided into three different groups: 185 individuals in the obstetricians and gynecologists' group, 10 in the plastic surgeons' group, and 125 in the other group.

## Participants characteristics

The mean age of the participants was 42.55 ± 10.56 years, with an average of 13.13 ± 10.24 years of professional experience. The majority of respondents were obstetricians and gynecologists (57.6%), followed by midwives (12.1%), general

practitioners (8.4%), plastic surgeons (3.1%), urologists (3.4%), psychiatrist (1.2%) and others (5.9%). Most of the respondents had a primary interest in women's health (46.7%) and the gynecology and obstetrics field (28.0%), followed by cosmetic medicine (11.8%) or sexual health (5.0%). The study revealed that nearly half of the participants worked in private hospitals and clinics (49.9%), while 23.1% were affiliated with university hospitals, and 27.1% practiced in both settings Table 1.

### Patients' interaction and training experience

Most participants, including 97.8% of obstetricians and gynecologists, 100% of plastic surgeons, and 88.8% of other specialists, reported being approached by patients with inquiries about FGCS, showing a significant difference (p = 0.002). The most common patient complaints were body image dissatisfaction, 65.4% among obstetricians and gynecologists,60% among plastic surgeons, and 27% among other specialties. Partner requests were another common concern, noted by 62.2% of obstetricians and gynecologists, 60% of plastic surgeons, and 43.7% of other specialties, and both were significantly higher among obstetricians and gynecologists compared to other groups (p=<0.001 and p = 0.005 respectively). Complaints related to sexual dysfunction were reported by 49.7% of obstetricians and gynecologists, which

**Table 1. Baseline characteristics of participants.**

| characteristics | Mean ± SD |
| --- | --- |
| **Age** | 42.55 ± 10.56 |
| **Experience (years)** | 13.13 ± 10.24 |
| | **Frequency (%)** |
| **Gender** | |
| Male | 39 (12.1) |
| Female | 26 (8.1) |
| **Work field** | |
| Obstetrician & Gynecologist | 185 (57.6) |
| Midwife | 39 (12.1) |
| Obstetrics & Gynecology resident | 26 (8.1) |
| Plastic surgeon | 10 (3.1) |
| Urologist | 11 (3.4) |
| Psychiatrist | 4 (1.2) |
| General practitioner | 27 (8.4) |
| Other | 18 (5.9) |
| **Filed of interest** | |
| Women health | 150 (46.7) |
| Gynecology & obstetrics | 90 (28.0) |
| Mental health | 15 (4.7) |
| Sexual health | 16 (5.0) |
| Cosmetic medicine | 38 (11.8) |
| Other | 12 (3.7) |
| **Work setting** | |
| University hospital | 74 (23.1) |
| Private hospital and clinic | 160 (49.9) |
| Both | 87 (27.1) |

Note: n: sample size. Data are presented as frequency and (percentages).

was also significantly different from other groups (p = 0.008). Hygiene issues were another problem that was also frequently reported (17.8% to 30% across groups) and was not different among three groups (p = 0.625).

Regarding the sources of information about FGCS, there were no significant differences among the three groups. Workshops were reported as the most common resource across all groups, with 57.8% of obstetricians and gynecologists, and other specialists and 50% of plastic surgeons participating in them, which was comparable among groups (p = 0.864). Conferences were another notable source, attended by 49.7% of obstetricians and gynecologists, 50% of plastic surgeons, and 57.9% of others (p = 0.357). Social media, websites, and interaction with other specialties were less common. obstetricians and gynecologists group performed FGCS more often than others (13.5% more than several times per month). The confidence levels in consulting patients varied among participants and by procedure. For instance, obstetricians and gynecologists expressed the highest confidence in performing labiaplasty (56.2% confident and 26.0% very confident). At the same time, procedures such as vulvar liposuction and vaginal rejuvenation saw lower confidence levels, with 76.4% and 58.3%, respectively, reporting no confidence. Plastic surgeons reported higher overall confidence, with 50% or more indicating confidence or high confidence for interventions such as fat or filler injections, clitoral hoodectomy, and vulvar liposuction. Among other specialties, the confidence level was generally lower, with over 50% reporting confidence in labiaplasty and perineoplasty but lower confidence for more specialized procedures like vaginal rejuvenation. Also, fewer than 10% reported very high confidence across most procedures. The difference between groups was statistically significant in all procedures except for clitoral hoodectomy (p = 0.17).

In terms of formal training in FGCS, the majority of participants across all groups (68.6% of obstetricians and gynecologists, 90% of plastic surgeons, and 75.2% of others) reported completing formal training, and differences were not significant (p = 0.196). courses on labiaplasty, monsplasty, and perineoplasty were the most common (61.1% of Obstetricians and Gynecologists, 50.0% of plastic surgeon and 52.0% of other groups, p = 0.255). Followed by filler and fat injection (p value = 0.003, 0.721, respectively). However, techniques such as vaginal thread lifting and laser-based procedures were less popular. Only a small proportion of participants (8.4%, n = 27) reported performing FGCS procedures on a regular basis, defined as several times per month. The majority had either limited or no hands-on experience with these surgeries, despite a relatively high percentage reporting formal training in FGCS techniques Table 2.

## Awareness of risks and ethical practices

Awareness of short-term risks was significantly different among groups (p- < 0.001), with the highest among obstetricians and gynecologists (62.0% aware) and the lowest among plastic surgeons (10.0% reporting little awareness and 90% reporting no familiarity). Still, the difference between groups was comparable (p = 0.499) in terms of long-term risk awareness, with 52.7% of obstetricians and gynecologists, 80% of plastic surgeons, and 52.4% of other specialties reporting total awareness. In contrast, only 4.9% of obstetricians and gynecologists and 6.3% of others reported no awareness. Ethical considerations were recognized by 54.3% of obstetricians and gynecologists and 51.2% of other specialties, but notably, all plastic surgeons (100%) acknowledged these ethical aspects, revealing significant differences between groups (p = 0.012). Most providers in the obstetricians and gynecologists, and plastic surgeons group discussed risks with patients (63.6% and 90%, respectively), while only 19.8% of other groups engaged in it, showing significant variation between groups (p = <0.001). Informed consent was obtained by 83.5% of obstetricians and gynecologists, 100% of plastic surgeons, and 91.1% of other specialties, showing significant variation among groups (p = <0.001) Table 3.

## Provider attitudes

A majority of respondents (56.8% of obstetricians and gynecologists, 60% of plastic surgeons, and 55.6% of others) agreed that all women should undergo consultation before FGCS (p = 0.952). Notably, 34.1% of obstetricians and gynecologists, 30.0% of plastic surgeons, and 35.7% of others believed that FGCS should not be performed on women under 18 years of age, which showed no significant variation between groups. Interestingly, a smaller comparable portion (8.6% of

**Table 2. Patient's interaction and clinical practice.**

| Question | Obstetricians and gynecologists (n = 185) | Plastic surgeons (n = 10) | Others (n = 125) | P value |
|---|---|---|---|---|
| **Have patients ever asked you about FGCS?** | | | | 0.002 |
| Yes | 181(97.8) | 10(100.0) | 111(88.8) | |
| No | 4(2.2) | 0(0.0) | 14(11.2) | |
| **What are the most common complaints from patients?** | | | | |
| Dyspareunia (painful intercourse) | 28(15.1) | 2(20) | 36(28.6) | 0.016 |
| Other sexual dysfunction (e.g., lack of satisfaction) | 92(49.7) | 2(20.0) | 43(34.1) | 0.008 |
| Vaginal discharge | 40(21.6) | 2(20) | 32(25.4) | 0.720 |
| Hygiene issues | 33(17.8) | 3(30.0) | 24(19.0) | 0.625 |
| Itching | 27(14.6) | 5(50.0) | 18(14.3) | 0.010 |
| Body image dissatisfaction | 121(65.4) | 6(60.0) | 34(27.0) | <0.001 |
| Partners request | 115(62.2) | 6(60.0) | 55(43.7) | 0.005 |
| **Source of information** | | | | |
| Internet(websites) | 56(30.3) | 4(40) | 26(20.6) | 0.107 |
| social media (Instagram,facebook, etcs) | 20(10.8) | 1(10) | 10(7.9) | 0.701 |
| Conferences | 92(49.7) | 5(50.0) | 73(57.9) | 0.357 |
| Other specialists | 62(33.5) | 7(70) | 42(33.3) | 0.057 |
| Workshops | 107(57.8) | 5(50.0) | 74(57.8) | 0.864 |
| **How often do you perform FGCS?** | | | | <0.001 |
| Several times per week | 13(7.0) | 0(0.0) | 2(1.6%) | |
| Several times per month | 12(6.5) | 0(0.0) | 0(0.0) | |
| rarely | 140(75.7) | 10(100.0) | 91(72.8) | |
| never | 20(10.8) | 0(0.0) | 32(25.6) | |
| **How confident do you feel in providing consultation for each of the following procedures?** | | | | |
| **Labiaplasty** | | | | <0.001 |
| Not confident | 33(17.8) | 0(0.0) | 37(29.6) | |
| Confident | 104(56.2) | 5(50.0) | 82(65.6) | |
| Very confident | 48(26.0) | 5(50.0) | 6(4.8) | |
| **Fat or filler injection to genitalia** | | | | <0.001 |
| Not confident | 104(58.4) | 0(0.0) | 51(40.8) | |
| Confident | 58(32.6) | 5(50.0) | 65(52.0) | |
| Very confident | 16(9.0) | 5(50.0) | 9(7.2) | |
| **Clitoral hoodectomy** | | | | 0.17 |
| Not confident | 100(56.2) | 6(60.0) | 58(46.4) | |
| Confident | 51(28.7) | 3(30.0) | 58(46.4) | |
| Very confident | 27(15.2) | 1(10.0) | 9(7.2) | |
| **Perineoplasty** | | | | 0.002 |
| Not confident | 49(26.9) | 3(30.0) | 52(41.6) | |
| Confident | 88(48.4) | 4(40.0) | 63(50.4) | |
| Very confident | 45(24.7) | 3(30.0) | 10(8.0) | |

*(Continued)*

**Table 2.** (Continued)

| Question | Obstetricians and gynecologists (n = 185) | Plastic surgeons (n = 10) | Others (n = 125) | P value |
|---|---|---|---|---|
| **Hymenoplasty** | | | | 0.005 |
| Not confident | 98(55.1) | 4(40.0) | 50(40.0) | |
| Confident | 59(33.1) | 3(30.0) | 65(52.0) | |
| Very confident | 21(11.8) | 3(30.0) | 10(8.0) | |
| **Vaginal re-juvenilizing** | | | | <0.001 |
| Not confident | 105(58.3) | 0(0.0) | 51(40.8) | |
| Confident | 0(0.0) | 4(40.0) | 64(51.2) | |
| Very confident | 22(12.2) | 6(60.0) | 10(8.0) | |
| **Vulvar liposuction** | | | | <0.001 |
| Not confident | 136(76.4) | 0(0.0) | 57(45.6) | |
| Confident | 30(16.9) | 5(50.0) | 63(50.4) | |
| Very confident | 11(6.2) | 5(50) | 5(4.00) | |
| **G-spot augmentation** | | | | <0.001 |
| Not confident | 12(67.0) | 5(50.0) | 51(40.8) | |
| Confident | 41(22.9) | 3(30.0) | 65(52.0) | |
| Very confidents | 18(10.1) | 2(20.0) | 9(7.2) | |
| **How comfortable do you feel consulting someone who requests FGCS?** | | | | |
| Not comfortable | 39(21.2) | 0(0.0) | 19(15.1) | 0.070 |
| Comfortable | 98(53.3) | 2(20.0) | 81(64.3) | |
| Very comfortable | 47(25.5) | 8(80.0) | 26(20.6) | |
| **Have you completed any formal training in FGCS?** | | | | 0.196 |
| Yes | 127(68.6) | 9(90.0) | 94(75.2) | |
| No | 58(31.4) | 1(10.0) | 31(24.8) | |
| **If yes, which courses have you attended? (Select all that apply):** | | | | |
| Labioplasty, monsplasty, perineoplasty | 113(61.1) | 5(50.0) | 65(52.0) | 0.255 |
| Filler injection | 67(36.2) | 9(90) | 45(36) | 0.003 |
| Fat injection | 69(37.3) | 5(50.0) | 48(38.4) | 0.721 |
| Vaginal threads lifting | 41(22.2) | 0(0.0) | 7(5.6) | <0.001 |
| Laser or radiofrequency | 51(27.6) | 4(40.00) | 21(16.8) | 0.043 |

Note: n: sample size. Data are presented as frequency and (percentages). P value ≤0.05 indicate statistical significance.

obstetricians and gynecologists, 0.0% of plastic surgeons, and 1.6% of others) found FGCS unacceptable solely for aesthetic reasons (p = 0.022). Less than half of all participants believed FGCS is a patient's personal choice (40.0% of obstetricians and gynecologists, 20.0% of plastic surgeons, and 35.5%of others), which showed comparable findings among groups (p = 0.119).

Regarding the prevalence of FGCS among cosmetic procedures, most participants believed that such surgeries constitute between 5% and 10% of the total, with 38.3% of obstetricians and gynecologists and 47.8% of other providers agreeing. The comparison highlighted significant differences (0.045).

**Table 3. Risk awareness and ethical practices.**

| Question | Obstetricians and gynecologists | Plastic surgeons | Others | P value |
|---|---|---|---|---|
| **Are you aware of the short-term risk of FGCS?** | | | | <0.001 |
| Yes | 114(62.0) | 0(0.0) | 41(32.8) | |
| A little | 60(32.6) | 1(10.0) | 78(62.4) | |
| no | 10(5.4) | 9(90.0) | 6(4.8) | |
| **Are you aware of the long-term risk of FGCS?** | | | | 0.499 |
| Yes | 97(52.7) | 8(80.0) | 66(52.4) | |
| A little | 78(42.4) | 2(20.0) | 52(41.3) | |
| no | 9(4.9) | 0(0.0) | 8(6.3) | |
| **Do you discuss the potential risks of FGCS with patients?** | | | | <0.001 |
| Yes | 110(63.6) | 9(90.0) | 21(19.8) | |
| no | 63(36.4) | 1(10) | 85(80.2) | |
| **Do you obtain informed consent from patients before procedures?** | | | | <0.001 |
| Yes | 152(83.5) | 10(100.0) | 113(91.1) | |
| No | 30(16.5) | 0(0.0) | 11(8.9) | |
| **Are you aware of the ethical considerations of FGCS?** | | | | 0.012 |
| Yes | 100(54.3) | 10(100.0) | 64(51.2) | |
| No | 84(45.7) | 0(0.0) | 61(48.8) | |

Note: n: sample size. Data are presented as frequency and (percentages). P value ≤0.05 indicate statistical significance.

Attitudes toward advertising also revealed a significant difference (<0.001): while only 29.2% of obstetricians and gynecologists endorsed advertising, this was notably higher among plastic surgeons (60.0%) and others (63.2%) Table 4.

## Discussion

This study evaluates the attitudes and knowledge of healthcare providers toward FGCS. FGCS refers to the surgical or non-surgical alteration of female genitalia, usually for aesthetic purposes. Our findings demonstrate that most of the participants, especially obstetricians and gynecologists, have been approached by patients regarding FGCS, with many of them complaining about body image dissatisfaction and partner requests. Consistent with our findings, Simonis et al. reported that nearly all general practitioners had an experience of visiting women with FGCS requests or concerns about genitalia normality [19]. This may be caused by various reasons. Insufficient women's knowledge of genital anatomy could lead to a misperception of normality. An international survey found that only about one-third of women were familiar with their vaginal appearance exactly [20]. Media exposure also plays a role. Sharp et al. Found that women seeking labiaplasty were more exposed to depictions of female genitalia on the Internet and in advertisements [21]. Similarly, Lowenstein et al. in 2013, found that using pornography content to obtain information about normal genitalia is more often is younger adults [22]. all these inappropriate and unreliable sources contribute to unrealistic expectations of genitalia appearance.

The cultural and religious context of Iran, as a conservative and predominantly Muslim society, plays a crucial role in shaping both patient and provider attitudes toward FGCS. In such settings, discussions around female sexuality and genital appearance may carry significant social stigma, and aesthetic genital surgeries can be seen as conflicting with traditional values around modesty and bodily integrity. This may explain the generally cautious approach observed among healthcare providers in our study, particularly regarding FGCS in adolescents and the use of advertising. In contrast, studies conducted in more liberal or secular societies, such as in Australia, the United States, or parts of Europe, report more permissive attitudes toward elective FGCS and a higher degree of normalization of these procedures in both clinical and

**Table 4.** Healthcare providers' attitudes.

| Question | Obstetricians and gynecologists (n = 185) | Plastic surgeons (n = 10) | Others (n = 125) | P value |
|---|---|---|---|---|
| Please indicate your opinion about FGCS (Select all that apply): | | | | |
| I have no opinion | 8(4.3) | 0(0.0) | 12(9.5) | 0.125 |
| FGCS is a personal choice if a woman wants it. | 74(40.0) | 2(20.0) | 38(35.5) | 0.119 |
| FGCS should not be performed on women under 18 years of age. | 63(34.1) | 3(30.0) | 45(35.7) | 0.911 |
| Consultation should be mandatory for all women before FGCS. | 105(56.8) | 6(60.0) | 70(55.6) | 0.952 |
| FGCS is unacceptable for aesthetic reasons alone. | 16(8.6) | 0(0.00) | 2(1.6) | 0.022 |
| It is no different from other types of cosmetic procedures | 35(18.9) | 0(0.0) | 34(27.0) | 0.057 |
| Do you know what percentage of cosmetic procedures are related to female genitalia? | | | | 0.045 |
| Less than 5% | 24(18.0) | 5(50) | 15(16.7) | |
| between 5 and 10 percent | 51(38.3) | 2(20.0) | 43(47.8) | |
| Between 10 and 40 percent | 45(33.8) | 3(30.0) | 29(32.2) | |
| Between 40 and 60 percent | 10(7.5) | 0(0.0) | 0(0.0) | |
| More than 60 percent | 3(2.3) | 0(0.0) | 3(3.3) | |
| Do you advertise for FGCS? | | | | <0.001 |
| Yes | 54(29.2) | 6(60.0) | 79(63.2) | |
| No | 131(70.8) | 4(40.0) | 46(36.8) | |

Note: n: sample size. Data are presented as frequency and (percentages). P value ≤0.05 indicate statistical significance.

media contexts [13,16,19]. These cross-cultural differences underscore the importance of considering sociocultural norms when evaluating FGCS practices and suggest that region-specific guidelines and ethical frameworks may be necessary.

According to our findings, the main sources of information and knowledge about FGCS are conferences and workshops. This is in line with the study by Sawan et al., which found conferences and online workshops to be the most popular sources [18].

Participants demonstrated varying confidence levels in performing FGCS, with obstetricians and gynecologists expressing the highest confidence in labiaplasty and the lowest in vulvar liposuction and G-spot augmentation. Plastic surgeons express overall more confidence in all aspects except G-spot augmentation. In contrast, among other groups, less than half of the participants express their confidence in most procedures. This disparity likely reflects unequal access to specialized training and the popularity of different types of FGCS as obstetricians and gynecologists in Iran perform most FGCS procedures, and labiaplasty and perineoplasty are more popular and requested. According to our results, over two-thirds of providers report training experience in FGCS, with labiaplasty, perineoplasty, and monsplasty as the most common training subjects. As in the Yegin et al. study, most physicians indicated that there was infrequently a medical justification to carry out other procedures such as hymenoplasty, G-spot augmentation, and clitoral hood reduction [17]. It is important to highlight that although over 70% of participants reported formal training in FGCS, only a small subset of the surveyed healthcare providers reported routinely performing FGCS (27 respondents reported conducting FGCS several times per month). This limited clinical experience may partially explain the overall variability in procedural confidence and risk awareness observed across specialties. Despite formal training, infrequent practice may result in gaps in practical knowledge and nuanced ethical decision-making. Future studies should consider stratifying attitudes and knowledge based on hands-on experience to better understand these discrepancies.

Gaps in training and awareness of long-term risks were evident across all groups. Most Plastic surgeons are unaware of short-term risks, while obstetricians and gynecologists and other groups report a higher level of awareness. This trend is not similar in long-term risk, where just a small portion of all participants categorize themselves as unaware. Over half

of respondents in obstetrics and gynecologists and other groups, as well as all plastic surgeons, believe they are familiar with the ethical considerations of FGCS; however, most participants obtain informed consent prior to procedures. In a guideline, it is recommended that the physician Explain FGCS procedures, including risks and complications, and lacking long-term data about outcomes [23]. In the study by Yegin et al., most participants believed that patient requests were enough to perform FGCS.

Healthcare attitude was comparable in most aspects. Most participants supported FGCS for aesthetic purposes but stressed the importance of pre-procedure counseling. The International Society for the Study of Vulvovaginal Disease" (ISSVD) highlighted the need for assessment of patient competency and recommended psychological counseling to all women who were seeking FGCS [13]. The need for considering mental health and relationship issues is also considered in the RACGP guideline, which recommends referral to psychiatrists or psychologists [23].

Interestingly, a minority of participants supported FGCS for patients under 18. Unlike our result, many other guidelines and studies believed FGCS should not be done for women under 18 [19,24].

Attitudes toward advertising also varied.

Most respondents in the obstetricians and gynecologists. group opposed advertising. However, more than half of the participants in other groups agreed. It may reflect the higher knowledge of normal genitalia and, to some extent, greater respect for women's genitalia differences. Yegin et al. also reported that about half of the participants disagreed with advertising FGCS to patients. Still, the opposite of our study was that most gynecologists thought it could be used [17].

## Strengths and limitations

This study's strengths include a large and diverse sample size along with the employment of validated tools for data collection, which enhances both reliability and generalizability. It is also the first study on this subject in Iran. Also, comparing different groups could be helpful in the future planning of training programs. Limitations include potential response bias due to the self-reported questionnaire. The insufficient number of plastic surgeons could also limit the generalizability of findings for this group. Moreover, it would be better to compare midwives' attitudes separately, as this group is often involved in FGCS, especially in their private office. It is also notable that although many participants reported formal training, only a minority were routinely involved in FGCS procedures. This highlights a critical gap between theoretical knowledge and practical experience, which should be addressed in future training initiatives and research designs. another limitation of this study is the lack of detailed socioeconomic data, which could have provided deeper insights into the findings.

## Future directions

Future research should explore how cultural, religious, and societal values influence provider attitudes and patient motivations in different regions. Comparative studies between conservative and liberal societies may offer deeper insights into how beliefs and norms shape FGCS practices. Furthermore, longitudinal studies are needed to assess the long-term outcomes and psychological impacts of FGCS on patients. Special attention should also be given to developing targeted training programs for providers, especially those with little clinical experience in FGCS. Finally, qualitative research involving both patients and providers could help uncover nuanced ethical dilemmas and guide culturally appropriate policy and educational development.

## Conclusion

In this study, we assessed the knowledge, attitudes, and practices of healthcare providers toward FGCS in a culturally conservative setting. While many respondents reported formal training and recognized ethical considerations, only a small number actively performed FGCS, and significant gaps in confidence and long-term risk awareness remain. Overall, attitudes toward offering FGCS for aesthetic purposes were cautiously positive, with an emphasis on the importance of pre-procedure counseling and informed consent. Our findings suggest the need for more comprehensive,

experience-based training and context-sensitive ethical guidance to better support both providers and patients in decision-making about FGCS.

## Author contributions

**Conceptualization:** Zinat Ghanbari, Marjan Ghaemi.

**Data curation:** Mohadese Dashtkoohi, Ideh Rokhzadi, Misa Naghdipour Mirsadeghi, Mahroo Rezaeinejad.

**Formal analysis:** Mohadese Dashtkoohi, Misa Naghdipour Mirsadeghi.

**Investigation:** Saeedeh Shirdel, Mahroo Rezaeinejad, Marjan Ghaemi.

**Methodology:** Mahroo Rezaeinejad.

**Project administration:** Zinat Ghanbari, Ideh Rokhzadi, Misa Naghdipour Mirsadeghi, Marjan Ghaemi.

**Resources:** Ideh Rokhzadi, Misa Naghdipour Mirsadeghi.

**Software:** Mohadese Dashtkoohi.

**Supervision:** Zinat Ghanbari, Marjan Ghaemi.

**Validation:** Zinat Ghanbari.

**Writing – original draft:** Saeedeh Shirdel.

**Writing – review & editing:** Mohadese Dashtkoohi.

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
