## [Decision Letter · Decision Letter 0]

21 May 2025

Dear Dr. Ghaemi,

Thank you for submitting your manuscript to PLOS ONE. After careful consideration, we feel that it has merit but does not fully meet PLOS ONE’s publication criteria as it currently stands. Therefore, we invite you to submit a revised version of the manuscript that addresses the points raised during the review process.

We look forward to receiving your revised manuscript.

Kind regards,

Federico Romano, M.D., Ph.D.

Academic Editor

PLOS ONE

2. In the online submission form, you indicated that [The data supporting this study's findings are available from the corresponding author upon reasonable request.].

Additional Editor Comments:

Dear Dr. Ghaemi,

I have read your article, "Attitude and Knowledge of Healthcare Providers Toward Female Genital Cosmetic Surgery: A Cross-Sectional Study." It is appealing due to the complex topic it addresses.

However, I must admit that a general revision of the English language is necessary to make the article more straightforward. I found many spelling mistakes that need to be corrected.

There is a significant disparity in the specialties of the participants. This should be addressed as it could lead to a different evaluation of the results.

Moreover, even if the objective of your article is to assess the knowledge of healthcare providers regarding FGCS, it is notable that only 27 participants admit to performing FGCS "several times". This must be emphasized in the article, and given the small number, it may be possible to focus on the results of these specific participants to assess whether any difference is present between those who perform this kind of surgery routinely and those who don't.

I find that the "attitude to advertising" part is not clear: what do you mean by "advertising"? Could you be more specific?

Please read carefully the attached reviewers' comments.

Reviewers' comments:

Reviewer's Responses to Questions

**Comments to the Author**

1. Is the manuscript technically sound, and do the data support the conclusions?

Reviewer #1: Yes

Reviewer #2: Partly

2. Has the statistical analysis been performed appropriately and rigorously?

Reviewer #1: Yes

Reviewer #2: Yes

3. Have the authors made all data underlying the findings in their manuscript fully available?

Reviewer #1: Yes

Reviewer #2: Yes

4. Is the manuscript presented in an intelligible fashion and written in standard English?

Reviewer #1: Yes

Reviewer #2: Yes

Reviewer #1: The authors present a very interesting manuscript regarding physicians' perspectives on FCGS from various specialties. The authors found the in general the physicians viewed FCGS in a positive light and would discuss short term complications associated with this surgery but without much thought on long-term complications.

This is a very difficult topic to study and discuss so I applaud the authors on trying to advance the field on this topic.

Were all these physicians from the same geographic area? Socioeconomic status?

The sociodemographic information of the studied physicians should definitely be included. Obviously, this topic is viewed vastly differently in different religions/cultures/geographic locations/etc. Therefore, this information is paramount to present regarding the participants of this study.

Furthermore, beyond just their specialty but the amount of training received specifically regarding FCGS should also be included in the study as this would inform the reader on how comfortable they would be discussing this topic. In addition, the number of previous FCGS procedures performed would be helpful to know and where their training regarding FCGS was performed.

Was this only in regards to cis-females or were trans-females also included? This could also skew the physicians' perspectives on this issue.

Reviewer #2: it is interesting to elaborate on the cultural aspects of a study made in a conservative religious environment and compare it to data in the literature. the conclusions and recommendations to the future need to be more clear

**Do you want your identity to be public for this peer review?** For information about this choice, including consent withdrawal, please see our Privacy Policy

Reviewer #1: No

Reviewer #2: No

---

## [Author Response · Author response to Decision Letter 1]

29 Jul 2025

Manuscript Title: Attitude and Knowledge of Healthcare Providers Toward Female Genital Cosmetic Surgery: A Cross-Sectional Study

We would like to thank the Academic Editor and the reviewers for their thoughtful comments and suggestions, which have helped us substantially improve our manuscript. Below, we provide point-by-point responses to each comment. Revisions have been made accordingly in the revised manuscript.

Editorial and Journal Requirements

1. English Language and Style

Editor: A general revision of the English language is necessary to make the article more straightforward. I found many spelling mistakes that need to be corrected.

Response:

Thank you for your observation. We have thoroughly revised the manuscript for spelling, grammar, and clarity. The revised version improves flow and ensures consistency in scientific language, capitalization, tense, and punctuation. A clean and tracked version has been uploaded.

2. Disparity in Specialties

There is a significant disparity in the specialties of the participants. This should be addressed as it could lead to a different evaluation of the results.

Response:

We agree. We have now explicitly acknowledged this as a limitation in the Discussion and Strengths and Limitations sections. We also clarified the composition of participant groups in the Results section.

3. Limited Number of FGCS Providers

Only 27 participants admit to performing FGCS "several times". This must be emphasized... consider focusing on these participants for subgroup analysis.

Response:

We thank the editor for this insight. We now emphasize the limited number of providers who routinely perform FGCS in the Results, Discussion, and Conclusion sections. Due to sample size constraints, subgroup analysis was limited, but we added a note suggesting future research explore this in greater detail.

4. Clarify “Advertising”

What do you mean by "advertising"? Could you be more specific?

Response:

We clarified this term in the Methods section. “Advertising” refers to public promotional activities such as online campaigns, clinic websites, and social media used by healthcare providers or clinics to promote FGCS services.

5. Ethics Statement Location

Your ethics statement should only appear in the Methods section.

Response:

Corrected. The ethics statement now appears only in the Materials and Methods section.

6. Data Availability

All data must be made freely available.

Response:

We have uploaded a de-identified version of the dataset as Supplementary File S1, in compliance with the journal’s data policy. We also updated the Data Availability Statement accordingly.

Reviewer #1

Were all these physicians from the same geographic area? Socioeconomic status?

Response:

We have clarified that all participants were based in Iran, and the majority practiced in urban healthcare settings. Unfortunately, detailed socioeconomic data were not collected, and this limitation is now acknowledged in the limitation section.

The sociodemographic information of the physicians should be included...

Response:

Thank you for your comment. Unfortunately, additional sociodemographic information for the physicians is not available in the dataset, as it was not collected during the study.

Amount of training received and prior FGCS performed should be reported.

Response:

These details were already partially reported. We now enhance this reporting in the Discussion, clearly stating the percentage with formal training and number of procedures performed

Was this study only about cis-females or did it include trans-females?

Response:

Thank you for this important point. We clarify in the Methods that the study focused solely on cisgender women. This clarification has been added explicitly.

It is interesting to elaborate on the cultural aspects of a study made in a conservative religious environment and compare it to data in the literature.

Response:

We appreciate this suggestion. We now expanded the Discussion to reflect the cultural and religious context of Iran and how this may affect healthcare providers’ attitudes. We also compared findings to those in more liberal or secular societies where FGCS is perceived differently.

Conclusions and recommendations to the future need to be clearer.

Response:

We revised the Conclusion and Future Directions sections to present clearer and more actionable recommendations, including calls for standardized training, ethical guidelines, and further subgroup analysis.

We hope these revisions meet the editorial and reviewer expectations and respectfully request reconsideration for publication. Thank you again for the opportunity to revise our work.

Sincerely,

Dr. Marjan Ghaemi, on behalf of all authors

---

## [Editor Report · Decision Letter 1]

26 Aug 2025

Dear Dr. Ghaemi,

Thank you for submitting your manuscript to PLOS ONE. After careful consideration, we feel that it has merit but does not fully meet PLOS ONE’s publication criteria as it currently stands. Therefore, we invite you to submit a revised version of the manuscript that addresses the points raised during the review process.

We look forward to receiving your revised manuscript.

Kind regards,

Federico Romano, M.D., Ph.D.

Academic Editor

PLOS ONE

Journal Requirements:

Additional Editor Comments:

Thank you for reviewing the article. The reviewers’ comments have helped make the paper more comprehensible and have highlighted both its strengths and its limitations. However, some spelling errors remain that must be corrected before publication. Please pay particular attention to the correct use of capital letters where required.

---

## [Author Response · Author response to Decision Letter 2]

9 Dec 2025

Dear Editor,

Thank you

I uploaded 3 files including response, highlighted file and clean file.

Manuscript Title: Attitude and Knowledge of Healthcare Providers Toward Female Genital Cosmetic Surgery: A Cross-Sectional Study

We would like to thank the Academic Editor and the reviewers for their thoughtful comments and suggestions, which have helped us substantially improve our manuscript. Below, we provide point-by-point responses to each comment. Revisions have been made accordingly in the revised manuscript.

Editorial and Journal Requirements

1. English Language and Style

Editor: A general revision of the English language is necessary to make the article more straightforward. I found many spelling mistakes that need to be corrected.

Response:

Thank you for your observation. We have thoroughly revised the manuscript for spelling, grammar, and clarity. The revised version improves flow and ensures consistency in scientific language, capitalization, tense, and punctuation. A clean and tracked version has been uploaded.

2. Disparity in Specialties

There is a significant disparity in the specialties of the participants. This should be addressed as it could lead to a different evaluation of the results.

Response:

We agree. We have now explicitly acknowledged this as a limitation in the Discussion and Strengths and Limitations sections. We also clarified the composition of participant groups in the Results section.

3. Limited Number of FGCS Providers

Only 27 participants admit to performing FGCS "several times". This must be emphasized... consider focusing on these participants for subgroup analysis.

Response:

We thank the editor for this insight. We now emphasize the limited number of providers who routinely perform FGCS in the Results, Discussion, and Conclusion sections. Due to sample size constraints, subgroup analysis was limited, but we added a note suggesting future research explore this in greater detail.

4. Clarify “Advertising”

What do you mean by "advertising"? Could you be more specific?

Response:

We clarified this term in the Methods section. “Advertising” refers to public promotional activities such as online campaigns, clinic websites, and social media used by healthcare providers or clinics to promote FGCS services.

5. Ethics Statement Location

Your ethics statement should only appear in the Methods section.

Response:

Corrected. The ethics statement now appears only in the Materials and Methods section.

6. Data Availability

All data must be made freely available.

Response:

We have uploaded a de-identified version of the dataset as Supplementary File S1, in compliance with the journal’s data policy. We also updated the Data Availability Statement accordingly.

Reviewer #1

Were all these physicians from the same geographic area? Socioeconomic status?

Response:

We have clarified that all participants were based in Iran, and the majority practiced in urban healthcare settings. Unfortunately, detailed socioeconomic data were not collected, and this limitation is now acknowledged in the limitation section.

The sociodemographic information of the physicians should be included...

Response:

Thank you for your comment. Unfortunately, additional sociodemographic information for the physicians is not available in the dataset, as it was not collected during the study.

Amount of training received and prior FGCS performed should be reported.

Response:

These details were already partially reported. We now enhance this reporting in the Discussion, clearly stating the percentage with formal training and number of procedures performed

Was this study only about cis-females or did it include trans-females?

Response:

Thank you for this important point. We clarify in the Methods that the study focused solely on cisgender women. This clarification has been added explicitly.

It is interesting to elaborate on the cultural aspects of a study made in a conservative religious environment and compare it to data in the literature.

Response:

We appreciate this suggestion. We now expanded the Discussion to reflect the cultural and religious context of Iran and how this may affect healthcare providers’ attitudes. We also compared findings to those in more liberal or secular societies where FGCS is perceived differently.

Conclusions and recommendations to the future need to be clearer.

Response:

We revised the Conclusion and Future Directions sections to present clearer and more actionable recommendations, including calls for standardized training, ethical guidelines, and further subgroup analysis.

We hope these revisions meet the editorial and reviewer expectations and respectfully request reconsideration for publication. Thank you again for the opportunity to revise our work.

Sincerely,

Dr. Marjan Ghaemi, on behalf of all authors

---

## [Editor Report · Decision Letter 2]

9 Feb 2026

Attitude and Knowledge of Healthcare Providers Towards Female Genital Cosmetic Surgery: A Cross-Sectional Study

PONE-D-25-11323R2

Dear Dr. Ghaemi,

We’re pleased to inform you that your manuscript has been judged scientifically suitable for publication and will be formally accepted for publication once it meets all outstanding technical requirements.

Kind regards,

Federico Romano, M.D., Ph.D.

Academic Editor

PLOS One

Additional Editor Comments (optional):

Thank you for submitting the revised version of your manuscript. We appreciate your careful attention to the reviewers’ and editors’ comments. I believe the article may open discussion on this underexplored topic and that it warrants further investigation.